# Detection of Local and Global Separability in Blurry Models as a Method for Explainable AI

## Abstract

The rapid advancement of deep learning and the deployment of intelligent systems in critical domains (medicine, finance, law) have brought the issue of decision interpretability to the forefront. Despite their high performance, "black box" architectures create a trust barrier for users due to the opacity of their internal structure. Existing post-hoc analysis methods (such as Grad-CAM, SHAP, and LIME) provide only local interpretation, failing to offer a global explanation of the system's logic.

Structural decomposition of complex systems into independent subsystems is considered a promising approach to addressing XAI challenges, as it enhances model controllability, verifiability, and safety. In this paper, this approach is implemented within the framework of blurry model theory — a methodology for logical formalization under conditions of incomplete and imprecise knowledge.

The authors extend the classical concept of a submodel to the class of blurry structures and introduce a mutual independence criterion: submodels are considered independent if the events they describe are stochastically independent. Based on this, the property of separability, i.e. the capacity of a blurry model to be decomposed into independent components is formalized. Criteria for local and global separability are investigated. A key result of the work is the proof of a theorem on the uniqueness of the "normal" (minimal) decomposition of a model.

## 1 Introduction

The current development of artificial intelligence technologies faces the problem of insufficient explainability of the decisions being made. This issue is particularly relevant in critical fields such as medicine, finance, and law. Traditional methods for creating deep neural networks are focused on high accuracy and efficiency, but their internal structure is perceived as a "black box". This makes it difficult for users to understand how the system makes decisions, which can lead to distrust and undesirable consequences.

To address this problem, approaches to Explainable Artificial Intelligence (XAI) are being developed, aimed at creating transparent systems Palchunov (2022a;b). These approaches help users understand how and why decisions were made while maintaining competitive recognition quality. For example, XAI includes visualization methods that show which data features influenced the model's conclusions.

Classical post-hoc methods, such as Grad-CAM, SHAP, or LIME, are considered as additional layers of interpretation for an already trained network. They analyze local predictions of the models and identify significant features for specific examples. However, this strategy improves local explainability and does not always take into account the need for a global explanation of the outputs **?**.

In many cases, a "black box" system consists of independent subsystems Chormai et al. (2024). It takes input data and distributes it across several branches, each of which solves its own part of the task. This allows the system to process information efficiently and reduces the probability of errors.

Detecting independent subsystems within a "black box" is a key task of XAI, as it is related to the manageability and safety of complex AI systems. Local explanations become meaningful at the subsystem level: for example, it is easier to explain the decision of a "credit scoring" or "medical module" than that of the entire monolith. Independent subsystems are easier to verify and to localize errors within. Given the growing attention to the ethical aspects of AI, the ability to explain a model's operation through independent subsystems can increase user trust in XAI technologies, which is important for ensuring fairness and transparency in decision-making **?**.

One of the most promising paths for the development of XAI is considered to be the design of hybrid systems, in which modern natural language processing models are complemented by mechanisms of symbolic Vityaev (2023) and probabilistic reasoning Goncharov (2025). Such an approach allows combining a broad knowledge base with the precision of formal logic, which ultimately guarantees both superior performance and a high degree of system reliability.

This work lies in the field of probabilistic reasoning and is devoted to research in the theory of blurry models, which is a methodology for the logical formalization of subject domains under conditions of inaccuracy and incompleteness of knowledge about these domains Yakhyaeva (2025a). The article considers the concept of a submodel of a blurry model as an extension of the concept of a submodel of a classical model, and also defines the mutual (pairwise) independence of submodels: submodels are independent if the events described in terms of one model are independent of the events described in terms of another model.

Based on the property of mutual independence of submodels of blurry models, the concept of the separability of a blurred model is introduced, i.e., its divisibility into mutually independent submodels. The criteria for the locally and globally of model separability are discussed. There can be several such decompositions for a specific model, among which a "minimal" decomposition, called normal, is selected; a theorem on the uniqueness of the normal decomposition is proven.

## 2 PAIRWISE INDEPENDENT SUBMODELS OF A BLURRY MODEL

In this paper, we will consider various blurry models of a fixed signature $\sigma$ containing no function symbols. Let $S(\sigma)$ denote the set of sentences of this signature.

Next, we will need to consider various subsets of the set of sentences $S(\sigma)$. Let us introduce the notation for these subsets:

$S_a(\sigma)$ – the set of all atomic sentences of the signature $\sigma$;

$S_p(\sigma)$ – the set of all positive conjuncts of the signature $\sigma$, i.e.

$$S_p(\sigma) = \{\psi \in S(\sigma) \mid \exists n \in N, \exists \varphi_1, ..., \varphi_n \in S_a(\sigma_A) : \psi = \varphi_1 \& ... \& \varphi_n, \};$$

$S_{qf}(\sigma)$ – the set of all quantifier-free sentences of the signature $\sigma$.

We will discuss the truth of formulas in a blurry model $\mathfrak{A}$. For convenience, to discuss the truth of sentences rather than arbitrary formulas in $\mathfrak{A}$, we expand the signature $\sigma$ with new constants. We will use the signature $\sigma_A \rightleftharpoons \sigma \cup \{c_a \mid a \in A\}$, where $\{c_a \mid a \in A\} \cap \sigma = \emptyset$. Furthermore, in $\mathfrak{A}$ it is true that $c_a^{\mathfrak{A}} = a$.

On the set of sentences $S(\sigma)$, let us introduce the semantic equivalence relation $\sim$ in the standard way. Let $\mathbb{S}(\sigma)$ denote the Lindenbaum-Tarski algebra $\langle S(\sigma)/_{\sim}; \vee, \&; , \neg, \top, \bot \rangle$.

**Definition 1** *Yakhyaeva G.E. (2023) The triple $\mathfrak{A}_\mu = \langle A, \sigma, \mu \rangle$ is called a **blurry model** if the valuation $\mu : S(\sigma_A) \rightarrow [0, 1]$ is a probability measure defined on the Lindenbaum-Tarski algebra $\mathbb{S}(\sigma_A)$.*

In defining the notion of a submodel of a blurry model, we require the following condition to hold: for any quantifier-free sentence of the extended signature of the submodel, its truth value in the submodel must coincide with its truth value in the model itself. To satisfy this condition, it is sufficient to require that this property holds for all positive conjuncts (see Yakhyaeva (2025b) ).

**Definition 2** *Yakhyaeva (2025b) Let $\mathfrak{A}_{\mu_A} = \langle A, \sigma, \mu_A \rangle$ and $\mathfrak{B}_{\mu_B} = \langle B, \sigma, \mu_B \rangle$ be blurry models of the same signature $\sigma$. We say that the blurry model $\mathfrak{B}_{\mu_B}$ is a **submodel** of the blurry model $\mathfrak{A}_{\mu_A}$ (denoted by $\mathfrak{B}_{\mu_B} \subseteq \mathfrak{A}_{\mu_A}$) if:*

*1) $B \subseteq A$;*

*2) for any constant $c \in \sigma$ we have $c^{\mathfrak{B}_{\mu_B}} = c^{\mathfrak{A}_{\mu_A}}$;*

*3) for any positive conjunct $\varphi \in S_p(\sigma_B)$ we have $\mu_B(\varphi) = \mu_A(\varphi)$.*

**Definition 3** *Yakhyaeva (2026) Consider blurry models $\mathfrak{A}_{\mu_A}, \mathfrak{B}_{\mu_B}, \mathfrak{C}_{\mu_C}$ such that $\mathfrak{B}_{\mu_B} \subseteq \mathfrak{A}_{\mu_A}$ and $\mathfrak{C}_{\mu_C} \subseteq \mathfrak{A}_{\mu_A}$. The models $\mathfrak{B}_{\mu_B}$ and $\mathfrak{C}_{\mu_C}$ will be called **pairwise independent** submodels of the model $\mathfrak{A}_{\mu_A}$ if for any positive conjuncts $\varphi \in S_p(\sigma_B)$ and $\psi \in S_p(\sigma_C)$ the following condition holds:*

$$\mu_A(\varphi \& \psi) = \mu_B(\varphi)\mu_C(\psi). \tag{1}$$

**Proposition 1** *If the models $\mathfrak{B}_{\mu_B}$ and $\mathfrak{C}_{\mu_C}$ are pairwise independent submodels of the model $\mathfrak{A}_{\mu_A}$, then the intersection of the models $\mathfrak{B}_{\mu_B}$ and $\mathfrak{C}_{\mu_C}$ is either empty or is a crisp model.*

*Proof.* Let us consider pairwise consistent models $\mathfrak{B}_{\mu_B}$ and $\mathfrak{C}_{\mu_C}$. Suppose that $B \cap C \neq \emptyset$. Then for any atomic sentence $\varphi \in S_a(\sigma_{B \cap C})$ we have

$$\mu_A(\varphi) = \mu_A(\varphi \,\&\, \varphi) = \mu_B(\varphi)\mu_C(\varphi) = \big(\mu_A(\varphi)\big)^2.$$

And this equality holds only if $\mu_A(\varphi) = 0$ or $\mu_A(\varphi) = 1$. Therefore, the submodel defined on the set $B \cap C$ is crisp.

**Corollary 1** *Any crisp submodels of a blurry model are pairwise independent in this model.*

**Corollary 2** *Any crisp submodel of a blurry model is pairwise independent from any other submodel of the given model.*

**Theorem 1** *Yakhyaeva (2026) Models $\mathfrak{B}_{\mu_B}$ and $\mathfrak{C}_{\mu_C}$ are pairwise independent submodels of the model $\mathfrak{A}_{\mu_A}$ if and only if condition (1) holds for any quantifier-free sentences of signatures $\sigma_B$ and $\sigma_C$.*

## 3 LOCALLY SEPARABLE MODELS

Consider the class $K_{submod}(\mathfrak{A}_{\mu_A}) = \{\mathfrak{B}_{\mu_B} \mid \mathfrak{B}_{\mu_B} \subseteq \mathfrak{A}_{\mu_A}\}$ of all submodels of the blurry model $\mathfrak{A}_{\mu_A}$. Obviously, (just as in the classical case) if the signature $\sigma$ of the model is purely relational, then any subset of the set $A$ A will form a submodel. If the signature $\sigma$ contains constant symbols, then the class $K_{submod}(\mathfrak{A}_{\mu_A})$ will contain the smallest submodel, which we will denote by $\mathfrak{A}_{\mu_{A_0}}$, where $A_0$ is the set of interpretations of the constants of the signature $\sigma$ in the model $\mathfrak{A}_{\mu_A}$.

**Definition 4** *A class $K \subseteq K_{submod}(\mathfrak{A}_{\mu_A})$ of submodels of the model $\mathfrak{A}_{\mu_A}$ is called a **cover** of the model $\mathfrak{A}_{\mu_A}$ if*

$$|\mathfrak{A}_{\mu_A}| = \bigcup_{\mathfrak{B}_{\mu_B} \in K} |\mathfrak{B}_{\mu_B}|.$$

A cover $K$ is called *degenerate* if $K = \{\mathfrak{A}_{\mu_A}\}$.

Obviously, the class $K_{submod}(\mathfrak{A}_{\mu_A})$ is a cover too.

**Definition 5** *We will say that a cover $K_1$ is **contained** in a cover $K_2$ (denoted by $K_1 \sqsubseteq K_2$) if for any submodel $\mathfrak{B}_{\mu_B} \in K_{submod}$ the following condition holds:*

$$\mathfrak{B}_{\mu_B} \in K_1 \Rightarrow \mathfrak{B}_{\mu_B} \in K_2.$$

*A cover $K$ will be called a **minimal cover** if there is no cover (distinct from $K$) contained in the cover $K$.*

**Proposition 2** *If a cover $K$ is minimal, then for any model $\mathfrak{B}_{\mu_B} \in K$ the class $K \backslash \mathfrak{B}_{\mu_B}$ is no longer a cover. $\mathfrak{A}_{\mu_A}$.*

*Proof.* Obviously, $K \backslash \mathfrak{B}_{\mu_B} \sqsubseteq K$. Then, if $K \backslash \mathfrak{B}_{\mu_B}$ is a cover, the class $K$ is not a minimal cover.

**Proposition 3** *For any cover $K$ there exists a minimal cover contained in this given cover.*

*Proof.* From the class $K$ we will gradually remove models that do not violate the property of being a cover. Then, by Proposition (2), we will obtain a minimal cover.

Note that the statement about the uniqueness of a minimal cover contained in a given cover is not true in the general case. To illustrate this, let us consider the model $\mathfrak{A}_{\mu_A}$ with the underlying set $A = \{a, b, c\}$ and a signature containing no constant symbols. Figure 1(a) shows a degenerate cover of this model. And Figure 1(b) depicts three different minimal covers of this model. That is, the submodels defined on the subsets $\{a, b\}, \{a, c\}$ and $\{b, c\}$ form a cover that is not minimal. By removing any of these three models, we will obtain a minimal cover.

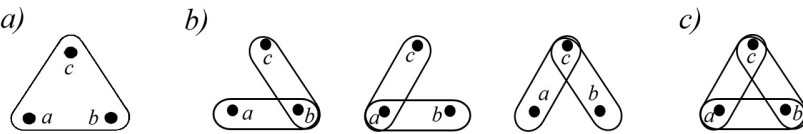

Figure 1: An example of different covers of a model whose underlying set consists of three elements and whose signature is purely predicative.

**Proposition 4** *If the cover $K$ is minimal, then for any models $\mathfrak{B}_{\mu_{B_1}}, \mathfrak{B}_{\mu_{B_2}} \in K$ we have $\mathfrak{B}_{\mu_{B_1}} \not\sqsubseteq \mathfrak{B}_{\mu_{B_2}}$.*

Note that the converse statement is not true. To illustrate this, one can also consider the example described above. The submodels defined on the subsets $\{a, b\}, \{a, c\}$ and $\{b, c\}$ on the one hand, are not submodels of each other, and on the other hand, form a cover that is not minimal. This cover is depicted in Figure 1(c).

**Definition 6** *A non-degenerate cover, all models of which are pairwise independent, will be called a **locally separable cover** of the $\mathfrak{A}_{\mu_A}$. A blurry model that has at least one locally separable cover will also be called a **locally separable model**; otherwise, the model will be called **locally entangled**.*

**Proposition 5** *Let $K$ be a locally separable cover and $K' \sqsubseteq K$. Then the cover $K'$ is also locally separable.*

**Proposition 6** *If the model $\mathfrak{A}_{\mu_A}$ is locally separable, then its smallest submodel $\mathfrak{A}_{\mu_{A_0}}$ is crisp.*

The proof follows directly from Proposition 1.

**Proposition 7** *If a model is locally separable, then it has at least two elements that do not belong to its smallest submodel.*

*Proof.* Let the model $\mathfrak{A}_{\mu_A}$ be locally separable. Then there exists a locally separable cover of it. Since the locally separable cover is not degenerate, it must contain at least two models that are not submodels of each other. For this, we must have at least two elements that do not belong to the intersection of the underlying sets of these models.

## 4 GLOBALLY SEPARABLE MODELS

Intelligent systems universally improve process efficiency, however, their continuous growth and increasing complexity make subsystem integration problems unpredictable even at the design stage. To maintain their optimal operation, it is crucial to identify and account for all mutual influences between components — both the easily noticeable explicit ones and the hard-to-distinguish implicit ones.

As an example, let us consider three submodels of the model $\mathfrak{A}_{\mu_A}$, namely $\mathfrak{B}_{\mu_{B_1}}, \mathfrak{B}_{\mu_{B_2}}, \mathfrak{B}_{\mu_{B_3}}$. Let them be pairwise independent. Then, according to Theorem 1, for any quantifier-free sentences $\varphi_i \in S_{qf}(\sigma_{B_i})$ we have

$$\mu_A(\varphi_1 \& \varphi_2) = \mu_{B_1}(\varphi_1)\mu_{B_2}(\varphi_2),$$
$$\mu_A(\varphi_1 \& \varphi_3) = \mu_{B_1}(\varphi_1)\mu_{B_3}(\varphi_3),$$
$$\mu_A(\varphi_2 \& \varphi_3) = \mu_{B_2}(\varphi_2)\mu_{B_3}(\varphi_3).$$

However, these conditions do not imply the fulfillment of the condition

$$\mu_A(\varphi_1 \& \varphi_2 \& \varphi_3) = \mu_{B_1}(\varphi_1)\mu_{B_2}(\varphi_2)\mu_{B_3}(\varphi_3).$$

That is, if we consider all three models together, we may lose the separability property. Thus, from local separability, we arrive at the concept of $k$-separability.

**Definition 7** *Consider the class of submodels $K \subseteq K_{submod}(\mathfrak{A}_{\mu_A})$. The class $K$ will be called $k$-separable (where $k \leq \|K\|$) if for any set of distinct models $\mathfrak{B}_{\mu_{B_1}}, ..., \mathfrak{B}_{\mu_{B_k}} \in K$ and for any positive conjuncts $\varphi_i \in S_p(\sigma_{B_i})$ the following condition holds:*

$$\mu_A\Big(\bigwedge_{i=1}^{k} \varphi_i\Big) = \prod_{i=1}^{k} \mu_{B_i}(\varphi_i). \tag{2}$$

Note that the fact that a cover is $k$-separable does not imply, in the general case, that it is locally separable. Moreover, Theorem 1 cannot be generalized to the case of $k$-separability. For this, we will need the concept of global separability.

**Definition 8** *A class of submodels $K \subseteq K_{submod}(\mathfrak{A}_{\mu_A})$ will be called **globally separable** if it is $k$-separable for any $k \leq \|K\|$. Otherwise, the class $K$ will be called **entangled**.*

Let us agree to consider any class of submodels to be 1-separable. Thus, a degenerate class (consisting of only a single model) will be globally separable.

**Theorem 2** *If a class of submodels $K \subseteq K_{submod}(\mathfrak{A}_{\mu_A})$ is globally separable, then condition (2) holds for any quantifier-free sentences of the corresponding signatures.*

*Proof.* Let the class $K$ be globally separable. Arbitrarily choose $n$ models $\mathfrak{B}_{\mu_{B_1}}, ..., \mathfrak{B}_{\mu_{B_n}} \in K$ (where $k \leq \|K\|$). We need to show that for any quantifier-free sentences $\varphi_i \in S_{qf}(\sigma_{B_i})$ condition (2) holds.

Consider the case where all the considered sentences $\varphi_i$ are conjunctions, i.e., they have the form

$$\varphi_i = \bigwedge_{j=1}^{k_i} \varphi_{ij} \ \& \ \bigwedge_{j=k_i+1}^{l_i} \neg\varphi_{ij},$$

where $\varphi_{ij} \in S_a(\sigma_{B_i})$ are atomic sentences of the corresponding signatures.

We will prove the identity (2) by induction on the number of negations occurring in the sentences $\varphi_1, ..., \varphi_n$. For the base case of the induction, consider the case where there is exactly one negation (in total across all sentences), i.e., for all $i = \overline{1, n-1}$ we have

$$\varphi_i = \bigwedge_{j=1}^{k_i} \varphi_{ij} \ and \ \varphi_n = \Big(\bigwedge_{j=1}^{k_n} \varphi_{nj}\Big) \& \neg\varphi_{nk_n+1}.$$

By the additivity of the measure $\mu$ we have

$$\mu_A\Big(\bigwedge_{i=1}^{n}\bigwedge_{j=1}^{k_i} \varphi_{ij}\Big) = \mu_A\Big(\Big(\bigwedge_{i=1}^{n}\bigwedge_{j=1}^{k_i} \varphi_{ij}\Big) \& \neg\varphi_{nk_n+1}\Big) + \mu_A\Big(\Big(\bigwedge_{i=1}^{n}\bigwedge_{j=1}^{k_i} \varphi_{ij}\Big) \& \varphi_{nk_n+1}\Big).$$

Therefore, from the global separability property, we obtain

$$\mu_A\Big(\bigwedge_{i=1}^{n} \varphi_i\Big) = \mu_A\Big(\Big(\bigwedge_{i=1}^{n}\bigwedge_{j=1}^{k_i} \varphi_{ij}\Big) \& \neg\varphi_{nk_n+1}\Big) =$$

$$= \mu_A\Big(\bigwedge_{i=1}^{n}\bigwedge_{j=1}^{k_i}\varphi_{ij}\Big) - \mu_A\Big(\Big(\bigwedge_{i=1}^{n}\bigwedge_{j=1}^{k_i}\varphi_{ij}\Big) \,\&\, \varphi_{nk_n+1}\Big) =$$

$$= \prod_{i=1}^{n}\mu_A\Big(\bigwedge_{j=1}^{k_i}\varphi_{ij}\Big) - \Big(\prod_{i=1}^{n-1}\mu_A\Big(\bigwedge_{j=1}^{k_i}\varphi_{ij}\Big)\Big)\cdot\mu_A\Big(\bigwedge_{j=1}^{k_n}\varphi_{nj}\,\&\,\varphi_{nk_n+1}\Big) =$$

$$= \prod_{i=1}^{n-1}\mu_A\Big(\bigwedge_{j=1}^{k_i}\varphi_{ij}\Big)\cdot\Big(\mu_A\Big(\bigwedge_{j=1}^{k_n}\varphi_{nj}\Big) - \mu_A\Big(\bigwedge_{j=1}^{k_n}\varphi_{nj}\,\&\,\varphi_{nk_n+1}\Big)\Big) =$$

$$= \Big(\prod_{i=1}^{n-1}\mu_{B_i}(\varphi_i)\Big)\cdot\Big(\mu_{B_n}\Big(\bigwedge_{j=1}^{k_n}\varphi_{nj}\,\&\,\neg\varphi_{nk_n+1}\Big)\Big) = \prod_{i=1}^{n}\mu_{B_i}(\varphi_i).$$

Thus, the base of the induction is proved. Further, the induction step can be shown similarly. Thereby, we have shown that the statement of the Theorem holds for any conjuncts of the corresponding signatures.

Let us now show that the Theorem is valid for any quantifier-free sentences as well. Consider the quantifier-free sentences $\varphi_i \in S_{qf}(\sigma_{B_i})$ (where $i = \overline{1,n}$).

We will prove this by induction on the number of sentences $n$. As the base of the induction, consider the case $n = 2$, i.e., consider the quantifier-free sentences $\varphi \in S_{qf}(\sigma_{B_1})$ and $\psi \in S_{qf}(\sigma_{B_2})$.

Since the measure $\mu$ is defined on the Lindenbaum-Tarski algebra, without loss of generality, we can assume that the sentences $\varphi$ and $\psi$ are in PDNF (Perfect Disjunctive Normal Form), i.e.,

$$\varphi = \varphi_1 \vee ... \vee \varphi_l, \ \psi = \psi_1 \vee ... \vee \psi_m,$$

where each $\varphi_i$ and $\psi_j$ is a conjunct of the signature $\sigma_{B_1}$ and $\sigma_{B_2}$ respectively.

Then, by the additivity property of the measure $\mu$, we obtain

$$\mu(\varphi\&\psi) = \mu((\varphi_1 \vee ... \vee \varphi_l)\&(\psi_1 \vee ... \vee \psi_m)) =$$

$$\mu\Big(\bigvee_{i=1}^{l}\bigvee_{j=1}^{m}\varphi_i\&\psi_j\Big) = \sum_{i=1}^{l}\sum_{j=1}^{m}\mu(\varphi_i\&\psi_j).$$

And since the condition of the Theorem has already been proved for the case of conjuncts, we obtain

$$\mu(\varphi\&\psi) = \sum_{i=1}^{l}\sum_{j=1}^{m}\mu(\varphi_i)\mu(\psi_j) = \Big(\sum_{i=1}^{l}\mu(\varphi_i)\Big)\cdot\Big(\sum_{j=1}^{m}\mu(\psi_j)\Big) =$$

$$= \mu\Big(\bigvee_{i=1}^{l}\mu(\varphi_i)\Big)\cdot\mu\Big(\bigvee_{j=1}^{m}\mu(\psi_j)\Big) = \mu(\varphi)\mu(\psi).$$

Further, the induction step can be shown similarly.

*The theorem is proved.*

**Definition 9** *A blurry model will be called a **globally separable model** if there exists at least one globally separable, minimal, non-degenerate cover $K$ of this model. Otherwise, the model will be called **entangled**.*

**Definition 10** *A cover $K$ of a blurry model $\mathfrak{A}_{\mu_A}$ will be called a **normal cover** of this model if the following properties hold:*

*a) The cover $K$ is globally separable;*

*b) The cover $K$ is minimal;*

*c) Each model $\{\mathfrak{B}_{\mu_B}\} \in K$ is entangled.*

**Theorem 3** *(on the uniqueness of a normal cover). Any blurry model has a unique normal cover.*

*Proof.* Any blurry model is either entangled or globally separable. Consider the case where the model $\mathfrak{A}_{\mu_A}$ is entangled.

Obviously, the degenerate cover $\{\mathfrak{A}_{\mu_A}\}$ is a normal cover of this model. We will show that it is unique.

Let the cover $K$ of the model $\mathfrak{A}_{\mu_A}$ be normal. Then, by Definition 10, it is globally separable and minimal. Therefore, by Definition 9, it must be degenerate. Thus, an entangled model has a unique normal cover, which is a degenerate cover.

Let us now consider the case where the model $\mathfrak{A}_{\mu_A}$ is globally separable.

Let $K_1$ and $K_2$ be normal covers of the model $\mathfrak{A}_{\mu_A}$. Let us fix a model $\mathfrak{B}_{\mu_B} \in K_1$ and consider the class of models
$$K' = \{\mathfrak{B}_{\mu_B} \cap \mathfrak{C}_{\mu_{C_i}} \mid \mathfrak{C}_{\mu_{C_i}} \in K_2\}.$$

Obviously, the class $K'$ is a cover of the model $\mathfrak{B}_{\mu_B}$. Moreover, due to the global separability of the class $K_2$, the class $K'$ is also globally separable.

However, the class $K'$ is not necessarily a minimal cover of the model $\mathfrak{B}_{\mu_B}$. Let us construct a class $K''$ such that $K'' \sqsubseteq K'$ and $K''$ is a minimal cover of the model $\mathfrak{B}_{\mu_B}$. According to Proposition 3, this is always possible. Given that global separability implies k -separability for every subclass of submodels, it trivially follows that the class $K''$ is also globally separable.

Is the class $K''$ non-degenerate? If the answer is yes, then the class $K''$ is a minimal, globally separable, non-degenerate cover, and therefore, by Definition 9, the model $\mathfrak{B}_{\mu_B}$ is globally separable. But this contradicts the condition that in a normal cover $K_1$ all models must be entangled.

Consequently, the class $K''$ is degenerate, i.e., it contains a single model. Thus, there exists a model $\mathfrak{C}_{\mu_{C_i}} \in K_2$ such that
$$K'' = \{\mathfrak{B}_{\mu_B} \cap \mathfrak{C}_{\mu_{C_i}}\}.$$

On the other hand, since the class $K''$ is a degenerate cover of the model $\mathfrak{B}_{\mu_B}$ we have
$$K'' = \{\mathfrak{B}_{\mu_B}\}.$$

Consequently, $\mathfrak{B}_{\mu_B} = \mathfrak{B}_{\mu_B} \cap \mathfrak{C}_{\mu_{C_i}}$. From this, we obtain that $\mathfrak{B}_{\mu_B} \subseteq \mathfrak{C}_{\mu_{C_i}}$.

Thus, due to the arbitrary choice of the model $\mathfrak{B}_{\mu_B}$, we obtain that for any model $\mathfrak{B}_{\mu_B} \in K_1$ there exists a model $\mathfrak{C}_{\mu_C} \in K_2$ such that $\mathfrak{B}_{\mu_B} \subseteq \mathfrak{C}_{\mu_C}$.

By similar reasoning, we also obtain that for any model $\mathfrak{C}_{\mu_C} \in K_2$ there exists a model $\mathfrak{B}_{\mu_{B'}} \in K_1$ such that $\mathfrak{C}_{\mu_C} \subseteq \mathfrak{B}_{\mu_{B'}}$.

Thus, we obtain that
$$\mathfrak{B}_{\mu_B} \subseteq \mathfrak{C}_{\mu_C} \subseteq \mathfrak{B}_{\mu_{B'}}.$$

Furthermore, since no minimal cover can contain a model together with its submodel (see Proposition 4), we obtain that
$$\mathfrak{B}_{\mu_B} = \mathfrak{C}_{\mu_C} = \mathfrak{B}_{\mu_{B'}}.$$

Thus, the classes $K_1$ and $K_2$ coincide.

*The theorem is proved.*

**Corollary 3** *A normal cover of a bluury model partitions the set of non-constant objects of this model into disjoint classes.*

*Proof.* Consider the blurry model $\mathfrak{A}_{\mu_A}$ and its normal cover $K$. Let $\mathfrak{A}_{\mu_{A_0}}$ be the smallest submodel of the model $\mathfrak{A}_{\mu_A}$. We need to show that the cover $K$ partitions the set $A \setminus A_0$ into disjoint classes.

We will prove this by contradiction. Suppose there exist submodels $\mathfrak{B}_{\mu_B}, \mathfrak{C}_{\mu_C} \in K$ such that $(B \cap C) \setminus A_0 \neq \emptyset$. Then, on the one hand, by Proposition 1, the model $\mathfrak{B}_{\mu_{B \cap C}}$ is a crisp model. On the other hand, it does not coincide with the smallest submodel, i.e., $\mathfrak{B}_{\mu_{B \cap C}} \neq \mathfrak{A}_{\mu_{A_0}}$.

Then, by Corollary 2, we obtain that the submodels $\mathfrak{B}_{\mu_{B \setminus C}}$ and $\mathfrak{B}_{\mu_{B \cap C}}$ are pairwise independent. Consequently, the class of models
$$\{\mathfrak{B}_{\mu_{B \setminus C}}, \mathfrak{B}_{\mu_{B \cap C}}\}$$

is a minimal, globally separable non-degenerate cover of the model $\mathfrak{B}_{\mu_B}$. Therefore, the model $\mathfrak{B}_{\mu_B}$ is not entangled, which contradicts the assumption that the cover $K$ is normal.

## 5 CONCLUSION

This paper is devoted to the mathematical formalization of the problem of detecting independent components of a complex system. This system is formalized as a blurry model of first-order logic, and its components are formalized as submodels of this model. It is shown that the property of local/global separability of components can be described formulaically. This will allow us to develop an algorithm for checking the existence of a partition of the system into independent components (i.e., checking for separability) of this system, as well as for finding a "normal" partition (see Theorem 3). Presumably, this will be a hierarchical algorithm that gradually constructs a partition of the set of system objects into disjoint classes (see Corollary 3).

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
