# OpenReview forum: "Detection of Local and Global Separability in Blurry Models as a Method for Explainable AI"
_mathai.club/MathAI/2026/Conference — 2026 Oral_

### Official Review · Reviewer_NLzY · 2026-03-11
**Detection of Local and Global Separability in Blurry Models**

**Rating:** 4
**Confidence:** 5

**Review:**

The article addresses the problem of interpreting the results of machine learning models. The authors focus on the difficulties of post-hoc interpretation, emphasizing the predominantly local nature of most such explanations. The paper proposes a formal theoretical framework based on an extension of the classical notion of a submodel to a class of so-called blurry models. Submodels are considered independent if the events described by the formulas of their signatures are stochastically independent. On this basis, several types of submodel coverings are introduced, and a theorem on the uniqueness of the so-called “normal” (minimal) decomposition of a model is formulated.

Among the positive aspects of the work is the attempt to formalize the problem of model interpretability using logical-algebraic constructions. The introduction of a probability measure on the Lindenbaum–Tarski algebra makes it possible to formulate properties of the model and its submodels in precise mathematical terms. The idea of describing model interpretability through the possibility of decomposing the model into independent components is also of interest. The paper presents a consistent system of definitions (submodels, coverings, local and global separability) and formulates several statements concerning the structure of such decompositions.

At the same time, the article contains several problems. First, the introductory section provides a somewhat simplified description of the current state of research in the field of interpretability. The authors contrast the proposed approach with post-hoc interpretation methods; however, the contemporary literature considers a considerably broader range of approaches. In addition to post-hoc explanations, there exist inherently interpretable models (for examples, Interpretable and generalizable graph learning via stochastic attention mechanism, ICML; Protgnn: Towards self-explaining graph neural networks, AAAI), methods for global interpretation of model behavior (Global Concept-Based Interpretability for Graph Neural Networks via Neuron Analysis, AAAI; Global Explainability of GNNs via Logic Combination of Learned Concepts, ICLR), and counterfactual explanation techniques (CMACE: CMAES‑based Counterfactual Explanations for Black‑box Models, IJCAI; Explaining and visualizing black‑box models through counterfactual paths, Pattern Analysis and Applications). In this respect, the characterization of existing approaches appears incomplete.

Furthermore, the connection between the proposed formalization and practical machine learning models remains insufficiently clarified. The independence of submodels is defined through exact factorization of the probabilities of formulas, which corresponds to full statistical independence. Such conditions are extremely strong and unlikely to hold in most real-world machine learning models, whose behavior is determined by complex dependencies among features and internal representations. As a result, the scope of applicability of the proposed theory remains unclear.

The mathematical part of the paper also contains certain inaccuracies and logical gaps. In particular, the proof of the theorem on the uniqueness of the normal covering relies on the claim that a minimal covering exists, derived from a procedure of sequentially removing elements. However, the validity of such an argument requires additional conditions (for example, finiteness of the considered class of submodels, which may be natural for machine learning models but still requires an explicit statement for mathematical rigor). In several places, independence is verified only for positive conjuncts, after which it is asserted that the result extends to all quantifier-free formulas. Such a step requires a proof of the closure of the corresponding algebras, which is only partially addressed in the text.

Additional issues arise in the use of the literature. In Chapter 1, the paper cites the article Interpreting Deep Neural Networks via Layerwise Feature Analysis, claiming that in many cases, a “black-box” system consists of independent subsystems. However, the cited work focuses on interpreting neural networks through the analysis of individual layers and does not draw conclusions about the existence of independent subsystems; the architecture under consideration remains a single integrated system. Thus, the reference appears to be used in a context that does not fully correspond to the content of the cited article. In addition, the manuscript does not distinguish between the terms interpretation and explanation, which are usually regarded as related but not identical concepts.

Small issues: several question marks appear in the first chapter, which creates the impression of incomplete or missing references. The text also mentions ideas related to widely used model interpretation methods without providing corresponding citations. In particular, references to commonly used methods such as Grad-CAM, SHAP, and LIME are absent.

Overall, the manuscript represents an interesting attempt to formalize model interpretability through a logical-probabilistic structure and the decomposition of a model into independent components. However, to strengthen both the scientific rigor and the practical relevance of the proposed approach, a more precise description of existing research directions, clarification of several formal definitions, and a more explicit justification of the applicability of the approach to modern machine learning models, many of which cannot naturally be represented as collections of independent submodels, would be beneficial.

---

### Official Review · Reviewer_6DuX · 2026-03-12
**The review of "Detection of Local and Global Separability in Blurry Models as a Method for Explainable AI"**

**Rating:** 6
**Confidence:** 4

**Review:**

This paper investigates the problem of identifying independent components of complex systems within the framework of explainable artificial intelligence. The authors develop a formal approach based on blurry model theory, where systems are represented as probabilistic logical structures and their components are modeled as submodels. The work introduces notions of pairwise independence of submodels and defines local, k-, and global separability of models. A key theoretical contribution is the proof of a theorem establishing the uniqueness of the normal decomposition (minimal globally separable cover) of a blurry model.

Strengths:
- The paper develops a mathematically rigorous framework for analyzing separability of components in complex systems.
- The notion of independence between submodels is clearly formalized using probability measures on logical formulas.
- The results on local and global separability provide a structured way to reason about decomposition of complex models.
- The uniqueness theorem for the normal cover offers an interesting theoretical guarantee for system decomposition.
- The work contributes to the theoretical foundations of explainable AI by connecting logical model theory with interpretability concepts.

Suggestions for improvement:

The paper could be further strengthened by:
- providing algorithmic procedures for detecting separability in practical systems;
- illustrating the framework with concrete examples or applications to AI models;
- clarifying the relationship between blurry models and modern machine learning architectures.

Final Recommendation:

POSTED / Poster-style acceptance with revision

Overall, the paper provides an interesting theoretical perspective on explainable AI based on logical model decomposition. The results may stimulate further research on formal methods for identifying independent subsystems in complex AI models.

---

### Official Review · Reviewer_PUJK · 2026-03-13
**The research analyzes AI decisions, focusing on issues of verification and interpretability of neural networks, proposing a logical-algebraic model of fuzzy formalization with component decomposition supported by Lindenbaum–Tarski calculations. The study requires refinement in providing more detailed substantiation of proofs and finalizing the authors' position.**

**Rating:** 6
**Confidence:** 4

**Review:**

This research examines the subsequent analysis of decisions made by artificial intelligence. It addresses issues such as the insufficient validation of obtained results and the complexity of understanding the internal structure of neural networks through which these decisions are made.

Quality: a formalized logical-algebraic model of fuzzy formalization for scientific domains is presented. A decomposition of the model's components for interpreting the posed problem is presented. Definitions describing the model's structure and the decomposition of its components are provided, along with statements substantiated by mathematical calculations.

Clarity: the research attempts to extend the classical submodel theory to variants of fuzzy models. The classical theory is described quite clearly, while there remains a need for a more detailed characterization of the proposed approach.

Originality: the originality of the study stems from the attempt to decompose the model into independent components, which is supported by mathematical calculations and a robust proof base for the proposed conclusions.

Pros: the theoretical provisions on model interpretability issues are substantiated by the Lindenbaum–Tarski algebraic model, with original calculations provided.

Cons: the research contains question marks, which suggests the incompleteness of the authors' position.

---

### Decision · Program_Chairs · 2026-03-14

**Decision:**

Accept (Oral)

**Comment:**

Dear Author(s),

On behalf of the Program Committee of the International Conference on Mathematics of Artificial Intelligence (MathAI 2026), we are pleased to inform you that your paper has been accepted for an oral presentation at MathAI 2026.

Your paper was evaluated through a rigorous two-stage review process involving both automated screening and expert review by members of the Program Committee. The reviewers recognized the quality and contribution of your work.

Presentation details:

- Format: Oral presentation (15–20 minutes + 5 minutes Q&A)
- Mode: You may present either in person (offline) at the conference venue in Sirius, Russia, or remotely via Zoom. Please indicate your preferred mode when confirming your participation.
- Conference dates: Marh 30 - April 3, 2026
- Website: https://mathai.club

Next steps:

1. Please confirm your participation and presentation mode by replying to this email mathai.club@yandex.ru no later than March 15, 2026 18:00 Moscow time.
2. If you plan to attend in person, the organizing committee will provide accommodation details separately.
3. Please prepare your final camera-ready manuscript according to the formatting guidelines available at https://mathai.club and upload it to OpenReview by March 15, 2026 18:00 Moscow time.

Should you have any questions regarding the program, logistics, or your presentation slot, please do not hesitate to contact us.

We look forward to your contribution to MathAI 2026.

With kind regards,

MathAI 2026 Program Committee
International Conference on Mathematics of Artificial Intelligence
https://mathai.club
OpenReview: https://openreview.net/group?id=mathai.club/MathAI/2026/Conference
Telegram: https://t.me/MathAI_club
Email: mathai.club@yandex.ru